# Inhibiting Methanogenesis in Rumen Batch Cultures Did Not Increase the Recovery of Metabolic Hydrogen in Microbial Amino Acids

**DOI:** 10.3390/microorganisms7050115

**Published:** 2019-04-27

**Authors:** Emilio M. Ungerfeld, M. Fernanda Aedo, Emilio D. Martínez, Marcelo Saldivia

**Affiliations:** 1Centro Regional de Investigación Carillanca, Instituto de Investigaciones Agropecuarias INIA, Temuco 4880000, Chile; aedo.enriquez13@gmail.com; 2Facultad de Ciencias Veterinarias, Universidad Austral de Chile, Valdivia 5090000, Chile; emiliomartinez@uach.cl (E.D.M.); marcelosaldiviamv@gmail.com (M.S.)

**Keywords:** rumen, fermentation, methane, inhibition, metabolic hydrogen, amino acids, microbial community composition

## Abstract

There is an interest in controlling rumen methanogenesis as an opportunity to both decrease the emissions of greenhouse gases and improve the energy efficiency of rumen fermentation. However, the effects of inhibiting rumen methanogenesis on fermentation are incompletely understood even in in vitro rumen cultures, as the recovery of metabolic hydrogen ([H]) in the main fermentation products consistently decreases with methanogenesis inhibition, evidencing the existence of unaccounted [H] sinks. We hypothesized that inhibiting methanogenesis in rumen batch cultures would redirect [H] towards microbial amino acids (AA) biosynthesis as an alternative [H] sink to methane (CH_4_). The objective of this experiment was to evaluate the effects of eight inhibitors of methanogenesis on digestion, fermentation and the production of microbial biomass and AA in rumen batch cultures growing on cellulose. Changes in the microbial community composition were also studied using denaturing gradient gel electrophoresis (DGGE). Inhibiting methanogenesis did not cause consistent changes in fermentation or the profile of AA, although the effects caused by the different inhibitors generally associated with the changes in the microbial community that they induced. Under the conditions of this experiment, inhibiting methanogenesis did not increase the importance of microbial AA synthesis as a [H] sink.

## 1. Introduction

Ruminants can transform roughages and non-protein nitrogen (N) otherwise unavailable to humans into meat, milk, wool, and traction. Essential to this nutritional flexibility is the presence of a complex microbial community in the rumen, which is able to degrade structural carbohydrates and synthesize amino acids (AA) by incorporating ammonia into carbon chains. In the fermentation of monosaccharides released from the degradation of complex carbohydrates in the rumen, metabolic hydrogen ([H]) is generated in redox reactions in bacterial, protozoal, and fungal cells, reducing intracellular cofactors. For fermentation to continue reduced cofactors need to be re-oxidized, which to an important extent occurs through the formation of dihydrogen (H_2_) [1,2]. Dihydrogen is a central intercellular intermediate in rumen fermentation, with hydrogenases-encoding genes being widespread in rumen bacteria and Archaea [3]. Most H_2_, along with other [H] donors such as formate, methanol, and methylamines is incorporated into methane (CH_4_) [2], which is the main electron sink in rumen fermentation.

Because CH_4_ formed in the rumen and released to the atmosphere constitutes a greenhouse gas [4] and also represents energy that is not incorporated into fermentation products that the animal can absorb and utilize [5], controlling the emissions of CH_4_ from ruminants can be regarded as an opportunity to simultaneously ameliorate climate change and improve animal productivity.

Various strategies to control CH_4_ formation in the rumen are currently being investigated. Rumen methanogenesis can be strongly inhibited by various chemical compounds [6] and oils such as linseed oil [7]. Whereas some of these additives and ingredients can inhibit methane production effectively, benefits in production have been inconsistent despite the theoretical gain of energy not lost as CH_4_ [8]. In vitro evidence from multiple experiments meta-analyzed has shown some undesirable and incompletely understood consequences of inhibiting methanogenesis, such as a decrease in total enthalpy output in volatile fatty acids (VFA) and a consistent decrease in the recovery of reducing equivalents pairs ([2H]) recovered in the main fermentation products [9]. It would be important to gain a thorough understanding of the changes occurring in [H] sinks when methanogenesis is inhibited in rumen fermentation.

It has been proposed that some of the unaccounted [2H] incorporated into the main fermentation products when methanogenesis is inhibited may be explained by increases in microbial biomass production, along with changes in its composition [9]. Previously, Hino and Russell [10] observed that inhibiting methanogenesis decreased deamination of AA in trypticase. Amination involves incorporation of one mol of [2H] per mol of ammonia, whereas deamination involves the release of [2H] [11]. Thus, part of the unaccounted [2H] resulting from methanogenesis inhibition could perhaps be explained by greater AA synthesis and lesser deamination. The main objective of this experiment was to study the changes in fermentation products and microbial AA as a consequence of inhibiting methanogenesis in rumen batch cultures using a range of chemical inhibitors and linseed oil. Our hypothesis was that changes in the synthesis of microbial AA could partly explain the decrease in [2H] recovery in main fermentation products when methanogenesis is inhibited. We also conducted a denaturing gradient gel electrophoresis (DGGE) to provide insights about how changes observed in fermentation and microbial growth associated with changes in the microbial community composition. Under the conditions of this experiment, uptake of [2H] for microbial AA biosynthesis did not explain the consistent decline occurring in [2H] recovery when methanogenesis is inhibited.

## 2. Materials and Methods

All procedures with animals employed in Project Fondecyt 1160764 were approved by Comité de Ética para el Uso de Animales en Investigación, Instituto de Investigaciones Agropecuarias INIA, Approval number 02/2016, on June 7 2016.

### 2.1. Treatments and Incubations

Approximately 0.5 L of rumen contents were sampled from the center of the rumen of each of two ruminally-cannulated non-pregnant, non-lactating, Holstein cows. Cows were fed ryegrass (*Lolium multiflorum*) hay [5.8% crude protein, 59.4% neutral detergent fiber (NDF), 6.0% total ashes, dry matter (DM) basis] once per day in the morning. Rumen contents were sampled before feeding around 10 am, and filtered through two layers of synthetic cloth. The fluid fraction from both cows was pooled, and the pooled fluid as well as the solids were transported to the laboratory in separate insulated containers.

One hundred milliliters of pooled rumen fluid were delivered under oxygen (O_2_)-free carbon dioxide (CO_2_) into a 500 mL Erlenmeyer flask, and approximately 50 mL of rumen solids from both cows were added. The resulting mixture of fluid and solids was subjected to an eggbeater at low speed under O_2_-free CO_2_ for 1 min (composed by bursts of approximately 3 s followed by 2 s breaks) to detach microorganisms adhered to solid particles. After egg beating, the mixture of fluid and solids was filtered through two layers of synthetic cloth under O_2_-free CO_2_ to obtain the rumen inoculum.

Subsequently, 130 µL of the resulting rumen inoculum was delivered under O_2_-free CO_2_ into 250 mL serum bottles containing 130 mL of the Goering and van Soest [12] medium including its reducing solution and with the following modifications: (i) Trypticase was not included; (ii) addition of 0.5 g/L yeast extract (Sigma-Aldrich 70161, St Louis, MO, USA); (iii) addition of 10 mL/L of Pfennigs mineral solution [13]; (iv) 10 mL/L of VFA solution [14]. Each bottle contained 800 mg cellulose (Sigma C6288, St Louis, MO, USA) of which its exact mass was recorded. Each bottle also contained one of the following: (i) 1 mL distilled water (Control); (ii) 235 µL of a 30 mM 2-bromoethanesulfonate (BES; Sigma-Aldrich 137502) solution; (iii) 1.3 mg anthraquinone (AQ; Sigma-Aldrich A90004) and 1 mL distilled water; (iv) 209 µL of a 0.05% (*v*/*v*) chloroform (CL; Merck 1.02445.2500, Darmstadt, Germany) solution and 791 µL distilled water; (v) 256 µL of a 0.05% (*v*/*v*) bromotrichloromethane (BTCM; Merck 8.01986.0100) solution and 744 µL distilled water; (vi) 40 µL of propynoic acid (propiolic acid; PA; Sigma-Aldrich P51400) and 1 mL distilled water; (vii) 75 µL of ethyl 2-butynoate (E2B; Sigma-Aldrich 425117) and 1 mL distilled water; viii) 163 µL linseed oil (LO; Nutra Andes Ltda., Valparaíso, Chile) and 1 mL distilled water; or (ix) 1 mL of a 1.3 M sodium nitrate (SN; Sigma-Aldrich S5506) solution. Final concentrations were chosen to target at least a 50% CH_4_ decrease relative to the control treatment and were 50 µM BES [15,16], 10 ppm AQ [17], 1.0 µM CL [18], 1.0 µM BTCM (concentration chosen equal to CL), 5 mM PA [19], 5 mM E2B [19], 1.25 mL/L LO, and 10 mM SN [20]. Bottles were immediately sealed and placed in a shaking water bath at 39 °C and 60 cycles/min. The initial gas pressure above ambient was determined using a pressure transducer (Sper Scientific 840065, Scottsdale, AZ, USA) after allowing the bottles to warm up for 10 min.

### 2.2. Sampling and Analytical Procedures

After 72 h of incubation, gas pressure above ambient was recorded, and a 12-mL gas sample was taken with a glass syringe and placed into vacutainers previously perfused with dinitrogen (N_2_) and evacuated to 0.2 atm. Bottles were then uncapped and final pH (Oakton^®^ pH 700 meter, Vernon Hills, IL, USA) and reducing potential (E*h*) (Schott Instruments BlueLine 31 Rx Ag/AgCl redox electrode in saturated KCl) immediately measured, and 0.5 mL of a 20% (*m*/*v*) sodium azide (Merck 6688) solution was added to arrest microbial activity [21]. One milliliter aliquots were transferred to eppendorfs containing 0.2 mL of 20% (*m*/*v*) m-phosphoric acid (Merck 1.00546.0500) or 1% (*v*/*v*) sulfuric acid, and kept at −20 °C until analyzed for VFA and ammonium ion (NH_4_^+^) concentration, respectively. Bottles were then shaken for a few seconds to re-suspend solid particles, and 1-mL aliquots were sampled using plastic Pasteur pipettes, and transferred to eppendorfs and stored at −80 °C until DNA was extracted for DGGE analysis. Bottles were again shaken for a few seconds to re-suspend solid particles and 10 mL were transferred to 15 mL Falcon tubes and kept frozen at −20 °C until lyophilized and analyzed for AA content. The remaining content of the bottles was centrifuged into pre-weighted bottles at 10,956× *g* and 6 °C for 20 min. The supernatant was discarded and the pellet was lyophilized. Tubes were then weighed, the pellet mass calculated by difference, and the pellet analyzed for DM, total ashes, total N, and NDF content [22].

Gas samples were analyzed for their content of CH_4_ and H_2_ in a Clarus 580 PerkinElmer GC equipped with a 60/80 Carboxen 1000 (Supelco, Bellefonte, PA, USA) packed column and a thermal conductivity detector and an isothermal oven temperature of 180 °C, using N_2_ at 30 mL/min as a carrier gas. Samples for VFA analysis were thawed, vortexed, and centrifuged at 16,000× *g* for 10 min. The supernatant was then filtered through 0.45 µm pore cellulose filters into 2 mL GC vials. One microliter VFA sample was injected in a PerkinElmer Clarus580 GC equipped with an Elite-FFAP (PerkinElmer, Shelton, CT, USA) capillary column and a flame ionization detector. Helium at 1.5 mL was the carrier gas. Initial temperature was 90 °C with a 12 °C/min ramp until 150 °C, which was held for 5 min.

Ammonium concentration was determined colorimetrically according to Kaplan [23]. Amino acids contents were analyzed in Falcon tubes containing 10-mL culture aliquots with suspended solid particles stored frozen. Tubes were lyophilized and subsamples of approximately 20 mg were transferred to hydrolysis tubes and added 1 mL of a 6 M hydrochloric acid and 1% (*m*/*v*) phenol solution. The tubes were then gassed with N_2_, closed air tightly, and incubated at 110 °C for 24 h [24]. Tubes contents were filtered through 0.45 µm cellulose filters and 20 µL of the filtrate diluted with ultrapure water to 500 µL. Ten microliters of the resulting dilution were derivatized with 6-aminoquinolyl-N-hydroxysuccinimidylcarbamate using the Waters^®^ AccQ·Tag^TM^ Amino Acid Analysis Method (Waters Corporation, Milford, MA, USA). Subsequently, 20 µL of derivatized sample were injected into a Hitachi L-7100 HPLC equipped with a C18 Sunshell column and a UV-Visible detector operating at 254 nm. Oven temperature was 36 °C using sodium acetate and 60% (*v*/*v*) acetonitrile as mobile phases. A standard AA mix containing all AA except Gln, Asn, and Trp at 100 µM concentration (Waters Corporation, Milford, MA, USA) was used to fit standard curves for each AA.

### 2.3. Denaturing Gradient Gel Electrophoresis Analysis

Frozen samples for DGGE analysis from the third incubation run of the experiment were thawed and total microbial gDNA was extracted using the repeated bead-beating protocol of Yu and Forster [25]. gDNA concentration was measured using a Maestrogen spectrophotometer (Maestrogen, Hsinchu City, Taiwan). The V3 region of the 16S rRNA bacterial gene was amplified by PCR using primers 341f-GC and 543r, following procedures described by Martínez et al. [26]. Purified PCR products were quantified using the NanoQuant Plate (Tecan Group Ltd., Männedorf, Switzerland), following the manufacturer’s instructions. Samples of two replicates per treatment from incubation run three were run on two DGGE gels. To standardize the amount of PCR amplicon loaded onto the gel, 100 ng of the purified bacterial PCR product was loaded into each well of a DGGE gel. PCR products were separated on an 8% polyacrylamide gel with a denaturing gradient of 30 to 60% urea/formamide in a solution of 0.5× TAE buffer at 60 °C using the DCode System (Bio-Rad, Hercules, CA, USA), and run at 100 V for 18 h. A ladder with multiples bands was run in two separate positions in the gel to aid in normalization during gel analysis. The gel was stained using the silver staining method [27]. Digitalized bacterial DGGE profiles were analyzed to calculate similarity matrices, and dendrograms with UPGMA algorithm for each gel based on band patterns with an optimization of 2% were constructed using Phoretix 1D (TotalLab Ltd., Newcastle upon Tyne, UK).

### 2.4. Calculations

Total gas pressure in the serum bottles at the beginning and at the end of the incubations was calculated as the measured gas pressure plus 101,325 Pa (1 atm). The number of moles of total gas present at the beginning and at the end of the incubations was calculated using the ideal gas law [28] considering a 120 mL headspace and the total gas pressure calculated as described above. Total gas production in moles was calculated by subtracting the initial from the final number of moles of total gas. The number of moles of CH_4_ and H_2_ present in the headspace were calculated by multiplying the number of moles of total gas at the end of the incubations by the analyzed volume percentage of each gas. The molar percentages of CH_4_ and H_2_ in total gas were then corrected and reported with respect to total gas actually produced. Because H_2_ is an intermediate in rumen fermentation with a large turnover that is incorporated into various fermentation pathways [3,29], H_2_ is reported as H_2_ accumulated, rather than as H_2_ production. Recorded E*h* was corrected to the Standard Hydrogen Electrode (SHE) by adding 197 mv [30].

Incubated cellulose contained 95% DM, and 0.03% total N, 97.7% NDF (DM basis), and undetectable total ashes. The amount of undigested cellulose substrate expressed as DM was calculated by dividing the NDF content in the undigested residue by cellulose NDF content. As incubated cellulose had undetectable total ash content, undigested cellulose DM was assumed to be equal to undigested cellulose organic matter (OM):Undigested cellulose (mg DM) = undigested cellulose (mg OM) = undigested pellet (mg DM) × (NDF% in undigested pellet ÷ 100) ÷ 0.977(1)

It was assumed that undigested cellulose had the same composition than the cellulose substrate. True digestibility of OM was calculated by dividing the difference between incubated OM in cellulose and undigested cellulose OM was calculated by Equation (1), expressing the result as a percentage.

OM true digestibility = [cellulose (mg OM) − undigested cellulose (mg OM)] × 100 ÷ cellulose (mg OM)(2)
Microbial OM production was calculated by subtracting the mass of undigested cellulose OM from the total mass of the undigested solid residue OM:Microbial biomass (mg OM) = undigested solid residue (mg OM) − undigested cellulose (mg OM)(3)

Net production of microbial N and AA were calculated by assuming that all N and AA present in the undigested residue corresponded to net microbial synthesis during the incubations, as their initial content was minimized by using pure cellulose as substrate and reducing the amount of inoculum from 20% (*v*/*v*) [12] to 0.1%.

Changes in [H] flows and sinks were quantified as pairs of reducing equivalents ([2H]) produced and incorporated. A balance of [2H] produced and incorporated in the catabolism of glucose (as the product of cellulose hydrolysis) to VFA and gases was calculated as by Ungerfeld [9] and Guyader et al. [21] (Appendix A), except that formate and heptanoate were not considered because they were not determined, and caproate was not considered for simplification, as there are two different pathways of formation with different implications to the electron balance [21].

Production and incorporation of [2H] was also calculated for the synthesis of AA from glucose, CO_2_ (from bicarbonate in the medium), VFA added to the medium, ATP carbon (in case of His), and NH_4_^+^ (Appendix A). Calculation of production and incorporation of [2H] for each AA was based on synthetic pathways previously described [1,11,31,32,33]. In some cases, the existence of more than one biosynthetic pathways for some AA can result in varying implications with regards to the uptake of disposal of [2H] [1,29]. For some AA, various biosynthetic pathways were therefore considered (Appendix A). In other cases, only the main pathway based on previous results with rumen microorganisms was considered. Transaminations were considered to indirectly incorporate 1 mol of [2H] per mol of NH_4_^+^ because amination in the synthesis of the AA donating NH_4_^+^ in the transamination reaction, generally glutamate or alanine, would incorporate 1 mol of [2H] [1]. Maximal and minimal net [2H] incorporation into AA synthesis was calculated for each experimental unit. Finally, overall maximal and minimal recovery of [2H] was calculated considering VFA and gases and AA synthetic pathways that maximized and minimized [2H] incorporation, respectively.

### 2.5. Statistical Analyses

There were two replicates per treatment per incubation and four incubation runs conducted on different days. All responses were modelled as a function of the fixed effect of the treatment (methanogenesis inhibitor) and the random effect of the incubation run. If the treatment effect was significant (*p* < 0.05), treatment means were separated using Tukey’s HSD.

Also, [2H] recovery in (i) VFA + gases; (ii) VFA + gases + AA calculated with maximal net incorporation of [2H] into AA, and (iii) VFA + gases + AA calculated with maximal net incorporation of [2H] into AA, were regressed against CH_4_ production per gram of OM incubated and truly digested.

Homogeneity of variances was evaluated by examining plots of residuals against predicted. We used residual normality plots to examine the assumption of normality of residuals. Outliers were identified as those treatment means whose absolute value studentized residuals were greater than *t*_68, 0.95_ (resulting from *t*_N-P-1, 0.95_, P being three parameters and N = 72 observations for a 95% confidence interval) and deleted from the analysis.

JMP^®^ 13.2.1 [34] was used for all statistical analyses.

## 3. Results

### 3.1. Fermentation

Inhibitors AQ, PA, and SN decreased (*p* < 0.05) gas production and total VFA concentration and increased (*p* < 0.05) final pH with respect to the control treatment (Table 1). All additives except for linseed oil inhibited (*p* < 0.05) CH_4_ production, and BES and AQ caused accumulation of H_2_ (*p* < 0.05). AQ decreased final E*h* (*p* < 0.05). Acetate molar percentage was increased (*p* < 0.05) by PA and SN, and propionate decreased (*p* < 0.05) by AQ, PA, E2B, and SN. Butyrate molar percentage was increased (*p* < 0.05) by AQ and E2B and unaffected by the other additives. The additives had no effect (*p* > 0.05) on isobutyrate or 2- and 3-methylbutyrate molar percentages or NH_4_^+^ concentration. Valerate molar percentage was increased (*p* < 0.05) by AQ, E2B and SN. Caproate molar percentage was increased (*p* < 0.05) by PA. The acetate to propionate molar ratio was increased by PA and SN.

### 3.2. Digestion and Microbial Biomass Production and Composition

True digestibility of OM was decreased (*p* < 0.05) by E2B (Table 2). There was no effect of the inhibitors on microbial OM production (*p* = 0.14), and PA and SN decreased (*p* < 0.001) microbial N production.

There was a general effect of treatments on microbial AA N, but no pair of means was found different by Tukey HSD contrasts (Table 2). The percentage of aspartate in total AA was increased (*p* < 0.05) by SN. The percentages of serine, arginine, threonine, tyrosine, and leucine in total AA were decreased (*p* < 0.05) by PA and SN. AQ and E2B increased (*p* < 0.05) the percentage of alanine, and PA of proline, in total AA. E2B decreased (*p* < 0.05) the percentage of phenylalanine in total AA. There were no effects of the inhibitors on the percentages of glutamate (*p* = 0.067), isoleucine (*p* = 0.058) or lysine (*p* = 0.17) in total AA.

### 3.3. Reducing Equivalents Balance

Production of [2H] in glucose fermentation to VFA and gases was decreased by PA and SN, whilst [2H] incorporation in VFA and gases and [2H] recovery were decreased by AQ, PA, E2B, and SN (Table 3; *p* < 0.05). Methanogenesis inhibition caused a pronounced decrease (*p* < 0.001) in [2H] recovery in VFA and gases (Figure 1 and Figure 2). There were no treatments effects on maximal (*p* = 0.34) or minimal (*p* = 0.16) net incorporation of [2H] into the synthesis of microbial AA. Inhibiting methanogenesis also decreased the overall recovery of [2H] considering the synthesis of microbial AA, whether net incorporation of [2H] into microbial AA was calculated to be maximal or minimal (Table 3 and Figure 1 and Figure 2). Even though considering the incorporation of [2H] into the synthesis of microbial AA increased [2H] recovery, the response of [2H] recovery to methanogenesis inhibition was not altered (Figure 1 and Figure 2).

### 3.4. Denaturing Gradient Gel Electrophoresis Analysis

There was variation among treatments in the DGGE patterns of bands (Figure 3). Bromotrichloromethane and chloroform clustered together. Bromoethanesulfonate showed a similar pattern of bands than the Control treatment. Sodium nitrate and AQ showed less intense band patterns than the Control and the other additives, with AQ forming a cluster by itself. A cluster of PA, E2B, and LO was also evidenced, with PA and E2B clustering closer.

## 4. Discussion

We used a chemically pure, N-depleted substrate such as pure cellulose to be able to quantify the production of microbial N and AA without a microbial marker. We decreased both microbial and non-microbial N initially present in the incubations by drastically decreasing the proportion of inoculum from 20 [12] to 0.1% (*v*/*v*). The rationale was that decreasing non-ammonium N initially present in the incubations would increase the proportion of N and AA in the incubation residue corresponding to microbial N, allowing to better evaluate the effects of the methanogenesis-inhibiting treatments. However, because such a pronounced decrease in the amount of inoculum would increase the lag time of the incubations substantially, the total incubation time was extended to 72 h based on preliminary results [35].

Fermentation was atypical in some aspects, such as propionate being a more important sink of [H] than CH_4_ in the Control treatment. The proportion of H_2_ in total gas in the Control treatment was also high considering that it was only surpassed by BES and AQ. Also, [H] recovery both in the control treatment and under methanogenesis inhibition was lower than what was reported in a previous meta-analysis of methanogenesis inhibited batch cultures [9]. The lower than typical importance of CH_4_ as a [H] sink compared with propionate and H_2_, and the relatively small acetate to propionate molar ratio for a cellulose substrate, suggest that some conditions in the Control treatment might have negatively affected the growth of methanogens [36]. It is possible that the small volume of inoculum, and perhaps the artificial nature of the pure cellulose substrate, resulted in deficiencies of vitamins or micronutrients which could have negatively affected methanogenesis, even though yeast extract and a complex micromineral solution were added to the medium intending to cover for possible deficiencies in minerals and vitamins.

All additives except for linseed oil inhibited CH_4_ production. Even though most additives also inhibited digestion and fermentation, as reflected by lower or numerically lower OM digestibility, total gas production, and total VFA concentration, CH_4_ production was not decreased solely on the basis of less OM being digested, as CH_4_ production per unit of truly digested OM was also decreased. A negative effect of linseed oil on CH_4_ production had been expected because the predominant fatty acid in linseed oil is the highly unsaturated α-linolenic acid [37], a strong inhibitor of methanogenesis in batch culture [15,38]. It is possible that the small amount of inoculum used in this experiment was insufficient for hydrolyzing triacylglycerides in linseed oil to release substantial amounts of α-linolenic acid during the incubations in order to exert its toxicity to methanogens [38].

Guyader et al. [20] reported a linear increase in acetate molar percentage in rumen batch cultures with increasing initial concentration of nitrate, although propionate molar percentage and the acetate to propionate molar ratio were not affected in their study. Nitrate has a dual mechanism to inhibit methanogenesis, by competing for [H] as a strong electron acceptor in its reduction to NH_4_^+^, and through its reduction intermediate nitrite, which is toxic to methanogens [39]. The observed increase in the acetate to propionate ratio and the numerical decrease in H_2_ concentration suggests that the reduction of nitrate to nitrite and NH_4_^+^ competed not only with CH_4_ but also with propionate as a [H] sink. In the present experiment, total reduction of nitrate to NH_4_^+^ would have potentially incorporated 5.2 mmol of [2H] and decreased CH_4_ production by 1.3 mmol (calculations not shown), which is greater than the observed decrease in CH_4_ production. However, complete reduction of nitrate to NH_4_^+^ did not seem to occur, as the final NH_4_^+^ concentration was not altered by SN. The unaffected final NH_4_^+^ concentration was not a consequence of increased NH_4_^+^ incorporation into microbial biomass, as SN resulted in lower microbial N production. Although nitrite concentration was not measured, it seems likely that much of added nitrate probably accumulated as nitrite, which could in turn explain the strong inhibition of fermentation reflected by the decrease in total VFA concentration in the SN treatment. Possibly, low availability of H_2_ in the SN treatment limited the reduction of nitrate to NH_4_^+^. Part of N in added nitrate might have also been incorporated into other compounds such as nitrous oxide [40].

Inhibiting methanogenesis is expected to cause a shift from acetate towards propionate production [36], which has been confirmed in a meta-analysis of methanogenesis-inhibited batch cultures [9]. Apart from SN, which, as discussed, acts as an alternative [H] sink to CH_4_, AQ, PA and E2B decreased CH_4_ but instead of increasing propionate molar percentage they decreased it. In previous work, PA and E2B at initial concentrations similar to the present experiment increased propionate production [19,41] although greater initial concentrations decreased propionate production [41,42]. García-López et al. [17] found increases or lack of effects of AQ at an initial concentration of up to 5 ppm in rumen batch cultures on propionate molar percentage, although 10 ppm in continuous cultures decreased propionate molar percentage. It seems that the greatest concentrations of AQ, PA, and E2B evaluated in the different studies may negatively affect succinate or propionate producers or both. In the present study, this might have been exacerbated by the small volume of inoculum, which could have decreased the initial microbial diversity in the incubations. The smallest final E*h* and greatest H_2_ concentration in total gas caused by AQ suggests that the system had problems to dispose [H] into pathways alternative to methanogenesis such as propionate formation.

The additives effects on digestion, fermentation, and microbial AA composition seemed to associate well with changes in the microbial community composition as reflected by the pattern of bands in the DGGE analysis. Chloroform and BTCM, both being chemical halogenated analogues of CH_4_, had similar effects on digestion, fermentation, and microbial AA composition, and clustered together in the DGGE analysis. Sodium nitrate and PA had similar effects on digestion and fermentation, inhibited the synthesis of the same AA, and also clustered together in the DGGE analysis. In previous work, however, PA and SN had clustered apart in a DGGE analysis [42]. The exception to the associations between the effects on digestion, fermentation, and the AA profile with the DGGE pattern of bands was linseed oil, which was largely inert to digestion and fermentation but clustered apart from the Control treatment. 

Inhibiting rumen methanogenesis in vitro has been found to steadily decrease the recovery of [2H] in the sum of main [H] sinks CH_4_, propionate, butyrate, and the intermediate H_2_, which typically accumulates when CH_4_ production is inhibited [9]. Ungerfeld [9] speculated that greater availability of [H] resulting from methanogenesis inhibition may stimulate the incorporation of NH_4_^+^ into carbon chains for AA synthesis. Under the concentrations of NH_4_^+^ generally found in the rumen, incorporation of NH_4_^+^ into carbon chains predominantly involves low-affinity systems such as NADP- and NAD-glutamate dehydrogenases, and alanine dehydrogenases [1], reactions that are coupled to the incorporation of [H] [11]. Contrary to our hypothesis, however, inhibiting methanogenesis did not increase the production of microbial OM or N, nor the incorporation of [2H] into the synthesis of microbial AA. Some inhibitors altered the proportion of particular AA in the total, but a general consequence of methanogenesis inhibition on the AA profile could not be established.

Taking into account microbial AA synthesis could increase [2H] recovery by between 1.0 and 5.7 percentage units (calculated as the differences in the intercepts in [2H] recovery in VFA and gases, and [2H] recovery in VFA and gases plus AA; Figure 1 and Figure 2). It is important to consider that in the microbial cultures of the present experiment, the incorporation of [H] into AA biosynthesis was maximized by not providing preformed AA. However, the increase in [2H] recovery due to considering the incorporation of [2H] into AA synthesis was independent of the extent of methanogenesis inhibition, as reflected by the similar slopes of the responses of [2H] recovery in VFA and gases, and in VFA and gases plus AA, to CH_4_ production (Figure 1 and Figure 2). Hence, at least under the conditions of this experiment, microbial AA biosynthesis did not contribute to explain the decrease in [2H] recovery caused by the inhibition of CH_4_ production. Future experiments should consider studying changes in other fermentation products and processes that can incorporate [H], such as formate, lactate, succinate, alcohols, and reductive acetogenesis, and reduced cell components such as long chain fatty acids [9].

## 5. Conclusions

Under the conditions of this experiment, biosynthesis of microbial AA accounted for a minor, but not negligible percentage of [2H] recovery, but it did not explain the decline in the recovery of [2H] in main fermentation products that occurs when methanogenesis is inhibited. In general, changes in fermentation and microbial biomass production and AA profile caused by eight different inhibitors of methanogenesis associated with the changes in the microbial community composition that they seemed to cause.

## Figures and Tables

**Figure 1 microorganisms-07-00115-f001:**
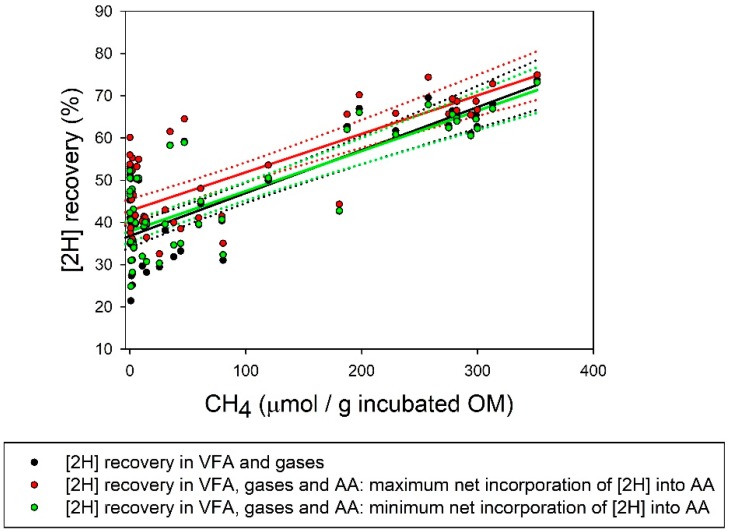
Response of recovery of reducing equivalents ([2H]) (%) to methane (CH_4_) produced per gram of organic matter (OM) incubated. Solid lines represent responses and dotted lines are 95% confidence bands. [2H] recovery (%) into: (i) Volatile fatty acids (VFA) and gases: • *y* = 36.9 (±1.24; *p* < 0.001) + incubation (random, *p* = 0.46) + 0.10 (±0.010; *p* < 0.001) *x*; *R*^2^ = 0.66; (ii) VFA, gases and amino acids (AA) with maximum incorporation of [2H] into AA synthesis: •
*y* = 42.6 (±2.11; *p* < 0.001) + incubation (random, *p* = 0.41) + 0.089 (±0.0095; *p* < 0.001) *x*; *R*^2^ = 0.69; (iii) VFA, gases and AA with minimum incorporation of [2H] into AA synthesis: •
*y* = 37.9 (±1.23; *p* < 0.001) + incubation (random, *p* = 0.80) + 0.095 (±0.0095; *p* < 0.001) *x*; *R*^2^ = 0.68.

**Figure 2 microorganisms-07-00115-f002:**
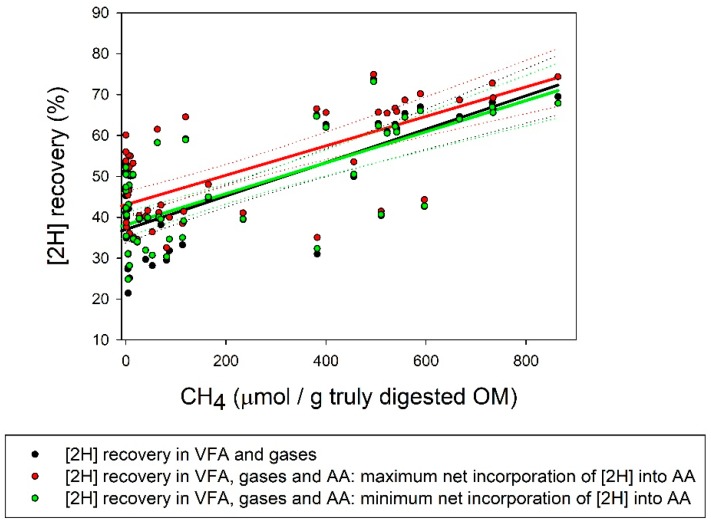
Response of recovery of reducing equivalents ([2H]) (%) to CH_4_ produced per gram of OM truly digested. Solid lines represent responses and dotted lines are 95% confidence bands. [2H] recovery (%) into: (i) VFA and gases: • *y* = 37.0 (±1.82; *p* < 0.001) + incubation (random, *p* = 0.77) + 0.041 (±0.0050; *p* < 0.001) *x*; *R*^2^ = 0.58; (ii) VFA, gases and amino acids (AA) with maximum incorporation of [2H] into AA synthesis: •
*y* = 42.9 (±2.83; *p* < 0.001) + incubation (random, *p* = 0.34) + 0.035 (±0.0044; *p* < 0.001) *x*; *R*^2^ = 0.62; (iii) VFA, gases and AA with minimum incorporation of [2H] into AA synthesis: •
*y* = 38.1 (±1.89; *p* < 0.001) + incubation (random, *p* = 0.60) + 0.038 (±0.0045; *p* < 0.001) *x*; *R*^2^ = 0.60.

**Figure 3 microorganisms-07-00115-f003:**
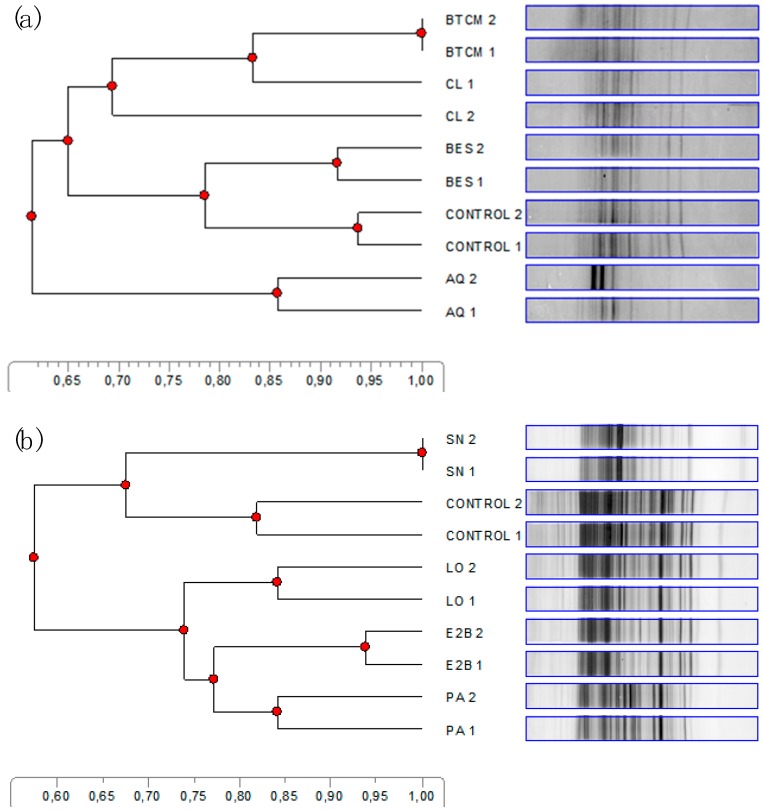
Dendrogram of bacterial denaturing gradient gel electrophoresis (DGGE) profile of the rumen in vitro batch cultures. (**a**) DGGE bacterial profiles of: BTCM, bromotrichlorometane; CL, chloroform; BES, 2-bromoethanesulfonate; Control, control treatment containing no inhibitor; AQ, 9,10-anthraquinone; (**b**) DGGE bacterial profiles of: SN, sodium nitrate; Control, control treatment containing no inhibitor; LO, linseed oil; E2B, ethyl-2-butynoate; PA, propynoic acid. Samples of two replicates per treatment of incubation run three were run on two DGGE gels.

**Table 1 microorganisms-07-00115-t001:** Effects of several inhibitors of methanogenesis on fermentation of rumen mixed batch cultures.

Response	Treatment										
	Control	BES ^1^	AQ	CL	BTCM	PA	E2B	LO	SN	SEM	*p* =
Gas production (mmol)	1.80 ^a3^	1.63 ^a^	0.91 ^cd^	1.67 ^a^	1.59 ^ab^	0.79 ^d^	1.24 ^bc^	1.71 ^a^	0.52 ^d^	0.17	<0.001
CH_4_ ^2^ (mol/100 mol total gas)	10.9 ^a^	0.55 ^bc^	2.34 ^bc^	1.36 ^bc^	2.70 ^bc^	0.076 ^bc^	3.13 ^b^	8.05^a^	ND ^4^	1.05	<0.001
CH_4_ production (µmol/g incubated OM)	242 ^a^	14.3 ^b^	29.3 ^b^	46.7 ^b^	51.1 ^b^	1.79 ^b^	62.1 ^b^	205 ^a^	ND	28.3	<0.001
CH_4_ production (µmol/g truly digested OM)	523 ^a^	30.4 ^bc^	121 ^bc^	68.3 ^bc^	110 ^bc^	9.26 ^c^	257 ^b^	575 ^a^	ND	57.5	<0.001
H_2_ accumulation (mol/100 mol total gas)	0.91 ^cd^	2.91 ^b^	5.60 ^a^	2.34 ^bc^	2.42 ^bc^	1.68 ^bcd^	0.36 ^d^	0.84 ^cd^	0.39 ^d^	0.64	<0.001
H_2_ accumulation (µmol/g incubated OM)	14.7 ^c^	60.2 ^a^	76.4 ^a^	51.2 ^ab^	52.3 ^a^	21.7 ^bc^	6.79 ^c^	19.5 ^c^	2.97 ^c^	9.95	<0.001
H_2_ accumulation (µmol/g truly digested OM)	37.4 ^bc^	155 ^abc^	307 ^a^	213 ^abc^	257 ^ab^	84.6 ^abc^	26.8 ^c^	68.3 ^bc^	8.52 ^c^	65.3	<0.001
Final pH	6.72 ^c^	6.70 ^c^	6.83 ^a^	6.72 ^c^	6.72 ^c^	6.79 ^ab^	6.76 ^bc^	6.70 ^c^	6.84 ^a^	0.042	<0.001
Final E*h*	−163 ^ab^	−167 ^ab^	−220 ^c^	−171 ^b^	−170 ^b^	−167 ^ab^	−165 ^ab^	−157 ^ab^	−140 ^a^	6.06	<0.001
Total VFA (mM)	27.2 ^a^	26.1 ^ab^	20.7 ^bc^	26.0 ^ab^	25.5 ^ab^	17.3 ^c^	23.7 ^ab^	27.2 ^a^	17.7 ^c^	7.95	<0.001
Acetate (mol/100 mol)	60.2 ^cd^	55.3 ^d^	62.3 ^c^	56.7 ^cd^	57.0 ^cd^	70.0 ^a^	63.2 ^bc^	57.8 ^cd^	69.7 ^ab^	2.18	<0.001
Propionate (mol/100 mol)	26.8 ^a^	30.0 ^a^	19.3 ^b^	29.5 ^a^	29.5 ^a^	15.8 ^b^	19.6 ^b^	29.4 ^a^	16.8 ^b^	2.06	<0.001
Butyrate (mol/100 mol)	8.04 ^c^	9.65 ^bc^	11.6 ^ab^	8.63 ^c^	8.55 ^c^	7.56 ^c^	12.7 ^a^	7.92 ^c^	7.13 ^c^	1.06	<0.001
Isobutyrate (mol/100 mol)	1.37	1.48	1.73	1.67	1.49	1.70	1.66	1.47	1.68	0.32	0.11
2- and 3-methylbutyrate (mol/100 mol)	1.86	1.75	2.31	1.77	1.63	1.88	2.53	1.88	1.92	1.02	0.081
Valerate (mol/100 mol)	1.30 ^c^	1.24 ^c^	1.80 ^a^	1.22 ^c^	1.32 ^c^	1.43 ^bc^	1.69 ^ab^	1.19 ^c^	1.78 ^a^	0.22	<0.001
Caproate (mol/100 mol)	0.44 ^b^	0.56 ^b^	0.88 ^ab^	0.50 ^b^	0.50 ^b^	1.75 ^a^	0.42 ^b^	0.43 ^b^	1.00 ^ab^	0.44	0.008
Acetate/propionate (mol/mol)	2.36 ^bcd^	1.96 ^d^	3.27 ^abc^	2.02 ^d^	2.04 ^cd^	4.49 ^a^	3.57 ^ab^	2.07 ^cd^	4.30 ^a^	0.30	<0.001
NH_4_^+^ (mM)	8.26	8.77	9.29	8.64	8.95	9.09	9.09	8.87	9.24	1.00	0.096

^1^ BES, 2-bromoethanesulfonate; AQ, 9, 10-anthraquinone; CL, chloroform; BTCM, bromotrichlorometane; PA, propynoic acid; E2B, ethyl-2-butynoate; LO, linseed oil; SN, sodium nitrate. ^2^ CH_4_, methane; H_2_, dihydrogen; E*h*, reducing potential; VFA, volatile fatty acids; NH_4_^+^, ammonium. ^3^ Unlike superscripts within the same row indicate significant (*p* < 0.05) differences according to Tukey HSD. ^4^ ND, not detected.

**Table 2 microorganisms-07-00115-t002:** Effects of several inhibitors of methanogenesis on digestion and microbial biomass production and composition of mixed rumen batch cultures.

Response	Treatment									SEM	*p* =
	Control	BES ^1^	AQ	CL	BTCM	PA	E2B	LO	SN		
True OM digestibility (%) ^2^	42.7 ^a 3^	41.4 ^ab^	31.2 ^ab^	46.4 ^a^	34.2 ^ab^	27.2 ^ab^	22.8 ^b^	31.4 ^ab^	31.2 ^ab^	6.13	0.008
Microbial OM (mg)	170	215	223	164	155	209	136	120	188	28.7	0.14
Microbial N (mg)	3.13 ^ab^	2.12 ^bc^	1.75 ^bc^	3.20 ^ab^	3.55 ^ab^	1.19 ^c^	1.97 ^bc^	3.97 ^a^	1.37 ^c^	0.47	<0.001
Total microbial AA-N (mg)	1.48	1.20	0.87	1.51	1.56	0.94	0.89	1.59	1.48	0.16	0.001
Amino acid (g/100 g total AA)											
Asp	11.7 ^bc^	12.8 ^ab^	5.73 ^c^	11.9 ^abc^	12.0 ^abc^	9.43 ^bc^	13.8 ^ab^	13.3 ^ab^	16.9 ^a^	1.23	<0.001
Glu	18.1	10.9	15.0	19.8	19.9	18.0	11.8	18.0	28.5	3.95	0.067
Ser	5.28 ^a^	5.55 ^a^	4.85 ^abc^	4.81 ^abc^	5.14 ^ab^	3.34 ^c^	6.12 ^a^	5.23 ^a^	3.53 ^bc^	0.42	<0.001
Gly	6.27 ^ab^	6.67 ^ab^	6.33 ^ab^	5.32 ^b^	6.14 ^ab^	7.17 ^ab^	8.11 ^a^	6.32 ^ab^	6.14 ^ab^	0.70	0.008
His	1.18 ^ab^	2.25 ^ab^	2.21 ^ab^	1.71 ^ab^	2.38 ^a^	0.62 ^b^	0.52 ^b^	1.47 ^ab^	0.73 ^ab^	0.47	0.005
Arg	4.26 ^ab^	4.98 ^a^	4.41 ^ab^	3.38 ^bcd^	4.26 ^abc^	2.82 ^cd^	3.23 ^bcd^	3.65 ^abc^	2.03 ^d^	0.48	<0.001
Thr	5.50 ^a^	5.77 ^a^	5.33 ^a^	4.97 ^a^	5.05 ^a^	2.86 ^b^	5.46 ^a^	4.94 ^a^	2.37 ^b^	0.43	<0.001
Ala	8.58 ^b^	10.2 ^ab^	13.8 ^a^	8.34 ^b^	8.42 ^b^	10.8 ^ab^	14.2 ^a^	10.1 ^ab^	8.93 ^b^	1.67	<0.001
Pro	3.26 ^bc^	3.83 ^abc^	4.46 ^ab^	3.93 ^abc^	3.61 ^abc^	4.74 ^a^	3.57 ^abc^	3.39 ^abc^	2.78 ^c^	0.36	0.002
Tyr	4.42 ^a^	4.57 ^a^	3.93 ^ab^	4.57 ^a^	3.75 ^ab^	1.83 ^b^	3.13 ^ab^	3.91 ^ab^	2.01 ^b^	0.60	<0.001
Val	6.03 ^a^	7.42 ^a^	8.13 ^a^	5.65 ^a^	6.09 ^a^	6.50 ^a^	7.33 ^a^	5.58 ^a^	5.82 ^a^	0.63	0.032
Ile	5.50	6.45	6.75	5.01	5.35	5.95	5.94	6.10	4.89	0.41	0.058
Leu	7.02 ^a^	7.95 ^a^	7.55 ^a^	6.14 ^ab^	6.54 ^a^	3.28 ^c^	7.75 ^a^	6.77 ^a^	3.95 ^bc^	0.66	<0.001
Lys	7.61	8.81	9.35	7.00	7.32	7.40	7.77	8.54	7.78	0.61	0.17
Phe	4.49 ^a^	4.71 ^a^	4.35 ^ab^	3.80 ^ab^	3.92 ^ab^	3.55 ^ab^	2.98 ^b^	3.98 ^ab^	3.57 ^ab^	0.37	0.009

^1^ BES, 2-bromoethanesulfonate; AQ, 9, 10-anthraquinone; CL, chloroform; BTCM, bromotrichlorometane; PA, propynoic acid; E2B, ethyl-2-butynoate; LO, linseed oil; SN, sodium nitrate; ^2^ OM, organic matter; N, nitrogen; AA, amino acids. ^3^ Unlike superscripts within the same row indicate significant (*p* < 0.05) differences according to Tukey HSD.

**Table 3 microorganisms-07-00115-t003:** Effects of several inhibitors of methanogenesis on the balance of reducing equivalents pairs ([2H]) in ruminal batch cultures.

Response	Treatment									SEM	*p* =
	Control	BES ^1^	AQ	CL	BTCM	PA	E2B	LO	SN		
**VFA and Gases**											
**[2H] Produced**											
Acetate (mmol)	4.17 ^a3^	3.79 ^ab^	3.33 ^bc^	3.81 ^ab^	3.76 ^ab^	2.99 ^c^	4.38 ^a^	4.14 ^a^	3.00 ^c^	1.27	<0.001
Propionate (mmol)	0.91 ^a^	0.97 ^a^	0.53 ^b^	0.95 ^a^	0.93 ^a^	0.38 ^b^	0.56 ^b^	1.02 ^a^	0.41 ^b^	0.21	<0.001
Butyrate (mmol)	1.28 ^b^	1.39 ^b^	1.26 ^bc^	1.27 ^b^	1.25 ^b^	0.81 ^c^	2.01 ^a^	1.35 ^b^	0.83 ^c^	0.53	<0.001
Valerate (mmol)	0.16 ^ab^	0.15 ^abc^	0.15 ^abc^	0.14 ^abc^	0.14 ^abc^	0.11 ^c^	0.18 ^a^	0.14 ^abc^	0.13 ^bc^	0.073	<0.001
Total [2H] produced (mmol) ^2^	6.52 ^ab^	6.30 ^ab^	5.27 ^bc^	6.17 ^ab^	6.08 ^ab^	4.28 ^c^	7.13 ^a^	6.66 ^ab^	4.38 ^c^	2.06	<0.001
[2H] incorporated											
Propionate (mmol)	1.81 ^a^	1.94 ^a^	1.07 ^b^	1.89 ^a^	1.86 ^a^	0.76 ^b^	1.12 ^b^	2.04 ^a^	0.82 ^b^	0.43	<0.001
Butyrate (mmol)	0.64 ^b^	0.70 ^b^	0.63 ^bc^	0.63 ^b^	0.62 ^b^	0.41 ^c^	1.01 ^a^	0.68 ^b^	0.42 ^c^	0.26	<0.001
Valerate (mmol)	0.22 ^ab^	0.20 ^abc^	0.20 ^abc^	0.19 ^abc^	0.18 ^abc^	0.14 ^c^	0.24 ^a^	0.19 ^abc^	0.18 ^bc^	0.097	<0.001
CH_4_ (mmol)	0.78 ^a^	0.037 ^b^	0.090 ^b^	0.15 ^b^	0.25 ^b^	0.0060 ^b^	0.26 ^b^	0.70 ^a^	ND ^3^	0.11	<0.001
H_2_ (mmol)	0.012 ^c^	0.046 ^a^	0.048 ^a^	0.041 ^a^	0.039 ^ab^	0.017 ^bc^	0.0082 ^c^	0.016 ^bc^	0.0024 ^c^	0.0082	<0.001
Total [2H] incorporated (mmol)	3.45 ^a^	2.91 ^ab^	2.09 ^bc^	2.85 ^ab^	2.97 ^ab^	1.33 ^c^	2.62 ^ab^	3.62 ^a^	1.35 ^c^	0.76	<0.001
[2H] recovery (%)	59.2 ^a^	48.9 ^ab^	40.1 ^bc^	47.1 ^ab^	49.5 ^ab^	29.2 ^c^	37.4 ^bc^	60.8 ^a^	27.1 ^c4^	3.72	<0.001
**Amino acids**											
Max [2H] net incorporation (mmol)	0.10	0.098	0.072	0.13	0.13	0.084	0.073	0.12	0.13	0.022	0.34
Min [2H] net incorporation (mmol)	−0.11	−0.097	−0.060	−0.13	−0.14	−0.091	−0.060	−0.13	−0.15	0.024	0.16
**Overall (VFA + gases + AA)**											
Max [2H] recovery (%)	62.0^a^	52.1 ^ab^	41.9 ^abc^	52.2 ^ab^	55.8 ^ab^	35.6 ^c^	38.1 ^bc^	63.3 ^a^	37.4 ^c4^	4.90	<0.001
Min [2H] recovery (%)	58.7^a^	49.1 ^ab^	40.1 ^abc^	47.5 ^ab^	51.3 ^ab^	31.0 ^c^	36.9 ^bc^	59.1 ^a^	29.3 ^c4^	4.13	<0.001

^1^ BES, 2-bromoethanesulfonate; AQ, 9, 10-anthraquinone; CL, chloroform; BTCM, bromotrichlorometane; PA, propynoic acid; E2B, ethyl-2-butynoate; LO, linseed oil; SN, sodium nitrate. ^2^ [2H], reducing equivalents pairs, CH_4_, methane; H_2_, dihydrogen; VFA, volatile fatty acids; AA, amino acids. ^3^ Unlike superscripts within the same row indicate significant (*p* < 0.05) differences according to Tukey HSD. ^3^ ND, not detected. ^4^ Probably underestimated as the reduction of nitrate to nitrite and ammonium was not determined.

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
