# Peer review of "Inhibiting Methanogenesis in Rumen Batch Cultures Did Not Increase the Recovery of Metabolic Hydrogen in Microbial Amino Acids"

_microorganisms, 2019, doi:10.3390/microorganisms7050115_

Round 1

Reviewer 1 Report

This manuscript reports results from an in vitro experiment assessing the impact of inhibiting methanogenesis on microbial amino acid balance.  The manuscript is generally well written and addresses an important topic relevant to the function and environmental impact of the rumen ecosystem.  The results were somewhat disappointing in that microbial amino acid synthesis was not found to be an important hydrogen sink during inhibition of methanogenesis.  Still, the finds are important and have merit.  There were some limitations in the current study that may warrant caution in over interpretation of the data and the authors were careful to include discussions on these limitations.  Overall, a worthwhile report.  I list below some specific comments for the authors' consideration.

Line 16 and line 43;  I think it is unfair to refer to ruminants as a "cause" of climate change.  Ruminants have long been contributors to the global carbon cycle and while their contemporary contribution is important it is arguably not drastically higher than when there were many, many ruminants roaming the earth's grasslands.  I think it is more fair to acknowledge that while ruminants should not be considered the cause of global warming they may indeed be part of the solution, is one can be achieved.  

Line 82, I am not sure what the plural of "thermos" is?  Perhaps it could be easily addressed by saying "insulated containers..."?

Line 84; should it say "Erlenmeyer flask..."?

Lines 332 to 344;  I am not convinced that all of the nitrate was metabolized completely to ammonia.  When I look at the total VFA produced by SN-treated cultures (17.7 mM), it seems as if fermentation was drastically inhibited.  If you have any sample remaining it might be worth measuring for nitrite.  I would suspect that H2 was limiting reduction of NO2 to ammonia. 

Author Response

Thank you for your comments, which we have incorporated in the revised version as follows:

Line 16 and line 43;  I think it is unfair to refer to ruminants as a "cause" of climate change.  Ruminants have long been contributors to the global carbon cycle and while their contemporary contribution is important it is arguably not drastically higher than when there were many, many ruminants roaming the earth's grasslands.  I think it is more fair to acknowledge that while ruminants should not be considered the cause of global warming they may indeed be part of the solution, is one can be achieved.  

We agree with the Reviewer and the intention was not to portrait ruminants as the culprits of climate change. Those sentences have been re-written to convey the sense of an opportunity to simultaneously decrease the emissions of greenhouse gases to the atmosphere and improve productivity (lines 16-18 and 47-50). We also added two sentences at the beginning of the Introduction explaining the important contributions of ruminants to mankind and how those are mediated, at least partly, by the rumen microbial community (lines 35-38).

Line 82, I am not sure what the plural of "thermos" is?  Perhaps it could be easily addressed by saying "insulated containers..."?

Corrected as suggested (line 88).

Line 84; should it say "Erlenmeyer flask..."?

Corrected as pointed out by the Reviewer (line 90).

Lines 332 to 344;  I am not convinced that all of the nitrate was metabolized completely to ammonia.  When I look at the total VFA produced by SN-treated cultures (17.7 mM), it seems as if fermentation was drastically inhibited.  If you have any sample remaining it might be worth measuring for nitrite.  I would suspect that H2 was limiting reduction of NO2 to ammonia. 

We agree with the Reviewer and have amended that section accordingly (lines 347-364). We currently do not have the technique set up to measure nitrate and nitrite, but speculate in the revised text that much of nitrate could have been reduced to nitrite and nitrite accumulated (lines 359-363). As far as nitrous oxide, we observed an unidentified peak in the gas chromatograms of the SN treatment, but our gas standards do not include nitrous oxide to confirm it (lines 363-364).

Reviewer 2 Report

The study is about inhibiting the methanogenesis and tracking metabolic [H] to find its sink. It seems like that [H] is not contributing to amino acid formation. The question is where does it go?

General comments-

1.     The order of abbreviations is confusing as, sometimes, the abbreviated form was written first and the full name was displayed later in the text. For an example “OM” was used in the beginning but Organic matter (OM) was written in line 270. There are many more (for example VFA, AA) which should be corrected.

2.     Lines 165-166, change to the “ideal gas law”  from the “general gas law”. Here is the example -(Singh et al., 2018 https://aem.asm.org/content/84/17/e00998-18)

3.     Please define what do you mean by the “gas production”?

4.     Acetate increase is the indication of fermentation leading to H2 synthesis however the increase was observed only with PA and SN. Why did not it increase with other treatments?

5.     Table-3, why is the % [2H] recovery not even 50%? Please write this in the discussion section.

Author Response

Thank you for your comments, which we have incorporated as follows:

The study is about inhibiting the methanogenesis and tracking metabolic [H] to find its sink. It seems like that [H] is not contributing to amino acid formation. The question is where does it go?

Amino acids biosynthesis did contribute to between 1.0 and 5.7% of [H] recovery, corresponding to the range of the differences in the intercept of the regression between [H] recovery calculated from VFA and gases only, or VFA, gases and amino acids. However, as the Reviewer points out, whilst amino acids biosynthesis can account for a minor proportion of [H] incorporation, it does not explain the steady decrease in [H] recovery occurring with methanogenesis inhibition (the slopes of [2H] recovery against methane production are similar whether or not amino acids biosynthesis is included). Thus, although [H] incorporation into amino acids formation does contribute a minor extent of [H] recovery, it does so to the same extent disregarding the inhibition of methanogenesis. Therefore, at least in this experiment, the biosynthesis of amino acids was not the answer to explain the decline of [H] recovery when methanogenesis is inhibited.

The important distinction between accounting for some [2H] incorporation and explain the decline in [2H] with methanogenesis inhibition was clarified (lines 402-409), and speculation about other alternative [H] sinks has been added (lines 411-414).

        1.  The order of abbreviations is confusing as, sometimes, the abbreviated form was written first  and the full name was displayed later in the text. For an example “OM” was used in the beginning but Organic matter (OM) was written in line 270. There are many more (for example VFA, AA) which should be corrected.

The order of the abbreviations has been checked and corrected. The definitions of the abbreviations was done once in the Abstract and then again in the main text, tables and figures (as per Instruction to Authors).

2.  Lines 165-166, change to the “ideal gas law”  from the “general gas law”. Here is the example -(Singh et al., 2018 https://aem.asm.org/content/84/17/e00998-18)

            Change made (line 173). We appreciate providing the reference.

3.      Please define what do you mean by the “gas production”?

            Gas production is defined mathematically in the text as the result of subtracting the number             of moles of total gas present at the beginning from the number of moles of total gas present             at the end of the incubation (lines 174-176). From a biological point of view, this variable              includes not only fermentation gas but carbon dioxide released from bicarbonate in the buffer.

4.      Acetate increase is the indication of fermentation leading to H2 synthesis however the increase was observed only with PA and SN. Why did not it increase with other treatments?

Because H2 is typically a fermentation intermediate with a very large turnover, one must distinguish between H2 production, which we actually do not typically measure, and H2 concentration or accumulation, which is what we actually report. In the rumen fermentation with functional methanogenesis, acetate production and H2 production are indeed positively associated. However, our measurements of H2 concentration or accumulation are not associated with acetate production, as H2 is rapidly incorporated into methanogenesis and its concentration is very low.

When methanogenesis is inhibited, H2 accumulates as methanogens grow slower. H2 accumulation thermodynamically causes a displacement of fermentation from acetate to propionate (Janssen, P. H. 2010. Anim. Feed Sci. Tech. 160:1-22.). Therefore, typically greater H2 concentration, or accumulation, (not production) is generally associated to less acetate and more propionate. In order to clarify this aspect in the manuscript, H2 appears now in Table 1 as “H2 accumulation”. This point is also explained in lines 179-181 in Materials and Methods.

Our measurements correspond of course to H2 concentration as a percentage of total gas and to H2 accumulation as number of moles released, rather than how much H2 was actually produced. The expectations were that inhibiting methanogenesis would result in increasing H2 accumulation due to slow growth of methanogens, and that that would shift the fermentation from acetate to propionate. An increase in acetate with methanogenesis inhibition was expected only for the case of SN, because as nitrate gets reduced to nitrite and ammonium it draws [H] away from not only methanogenesis but also from propionate production, whereas H2 utilization in nitrate reduction thermodynamically favors acetate production. Why PA increased acetate is more difficult to understand. It might have somehow stimulated reductive acetogenesis, although this might be too speculative for the results we have so as to include it in the Discussion.

5.      Table-3, why is the % [2H] recovery not even 50%? Please write this in the discussion section.

This aspect has been added to the second paragraph of the Discussion, in which some atypical aspects of fermentation in this study are described and discussed (lines 328-330).